# Calpains as Potential Therapeutic Targets for Myocardial Hypertrophy

**DOI:** 10.3390/ijms23084103

**Published:** 2022-04-07

**Authors:** David Aluja, Sara Delgado-Tomás, Marisol Ruiz-Meana, José A. Barrabés, Javier Inserte

**Affiliations:** 1Cardiovascular Diseases Research Group, Vall d’Hebron Institut de Recerca (VHIR), Vall d’Hebron Hospital Universitari, Passeig Vall d’Hebron 119-129, 08035 Barcelona, Spain; david.aluja@vhir.org (D.A.); sara.delgado@vhir.org (S.D.-T.); marisol.ruizmeana@vhir.org (M.R.-M.); jabarrabes@vhebron.net (J.A.B.); 2Centro de Investigación en Red de Enfermedades Cardiovasculares (CIBERCV), 28029 Madrid, Spain

**Keywords:** calpain, calpastatin, myocardial hypertrophy, heart failure

## Abstract

Despite advances in its treatment, heart failure remains a major cause of morbidity and mortality, evidencing an urgent need for novel mechanism-based targets and strategies. Myocardial hypertrophy, caused by a wide variety of chronic stress stimuli, represents an independent risk factor for the development of heart failure, and its prevention constitutes a clinical objective. Recent studies performed in preclinical animal models support the contribution of the Ca^2+^-dependent cysteine proteases calpains in regulating the hypertrophic process and highlight the feasibility of their long-term inhibition as a pharmacological strategy. In this review, we discuss the existing evidence implicating calpains in the development of cardiac hypertrophy, as well as the latest advances in unraveling the underlying mechanisms. Finally, we provide an updated overview of calpain inhibitors that have been explored in preclinical models of cardiac hypertrophy and the progress made in developing new compounds that may serve for testing the efficacy of calpain inhibition in the treatment of pathological cardiac hypertrophy.

## 1. Introduction

Despite significant advances in therapeutic management, heart failure (HF) is still a major public health problem, with high prevalence, poor clinical outcomes, and large healthcare costs [1]. Cardiac hypertrophy, defined as the absolute increase in ventricular mass, is among the most robust markers of increased risk for developing HF, independently of the underlying cause [2,3]. It is proposed that hypertrophy initially develops as an adaptive response to increased biomechanical stress, serving to minimize wall stress and maintain contractile function. However, sustained hypertrophic stimulation in the setting of a disease results in the disruption of this physiological adaptation to stressors and evolves into a progressive development of pathological or maladaptive hypertrophy, triggering the transition to a state of decompensation and clinical HF [4].

Cardiac hypertrophy as a clinical entity is observed in a broad range of pathologies linked to sustained pressure and volume overload, as well as ischemic disease and genetic disorders [5]. From a population standpoint, the most common cause of hypertrophy is systemic hypertension, which is considered the single most important predictor of ventricular hypertrophy, but it is also related to a number of other conditions, including myocardial infarction, aortic stenosis, and regurgitant valvular heart disease. Other common comorbidities, such as obesity, diabetes, or hypercholesterolemia, may play a synergistic and potentially independent role in the development of hypertrophy.

Despite some discrepancies [6], preclinical and epidemiological studies indicate that the prevention of hypertrophy constitutes a meaningful clinical objective [7,8,9,10,11]. Therefore, a greater understanding of the molecular mechanisms governing the development of hypertrophy and the transition from adaptive to pathologic hypertrophy would help to identify novel targets and strategies aimed at preventing its progression to HF [12].

Hypertrophic transformation of the cardiomyocyte is a complex process that involves the alteration of multiple cellular mechanisms controlling both protein synthesis and degradation. On the one hand, enhanced protein synthesis results from a wide range of transcriptional and post-transcriptional events, with the activation of a pattern of gene expression reminiscent of that observed during fetal development being the most relevant (“fetal gene program”) [13,14,15,16,17]. On the other hand, variations in the rate of protein degradation affect the activity of the main cellular proteolytic systems: ubiquitin proteasome, autophagy, and the calpain/calpastatin system.

Hypertrophy has been largely associated with aberrant ubiquitin-proteasome system (UPS) activity resulting from changes in the expression of ubiquitin ligases and deubiquitinating enzymes or abnormal UPS post-translational modifications [18]. However, the literature analyzing the role of these changes in the development of hypertrophy is contradictory. Ablation of ubiquitin ligases has been associated with the attenuation of cardiomyocyte hypertrophy and diastolic dysfunction [19,20,21] and also with more severe hypertrophy [22,23,24]. Similarly, while in some studies, pharmacological inhibition of proteasome components resulted in the regression or prevention of cardiomyocyte hypertrophy [25,26,27], others have shown that the chronic treatment with proteasome inhibitors caused hypertrophy and HF under baseline conditions [28]. Thus, it remains unclear whether the inhibition of the UPS is protective or detrimental in the context of cardiac hypertrophy.

The contribution of autophagy to cardiac hypertrophy has recently been reviewed [29,30]. Autophagy induction has been observed in several pathological conditions, including hypoxia, endoplasmic reticulum stress, oxidative stress, and nutrient starvation [31], and was initially suggested to be also enhanced in models of hypertrophy induced by pressure overload [29,32]. However, recent reports support that myocardial autophagic activity is depressed during the progression from adaptative cardiac hypertrophy to HF [33,34,35,36]. This autophagy insufficiency has been proposed to be caused by either inadequate autophagosome formation or impaired autophagosome clearance [29,37]. From these studies, it is suggested that those strategies enhancing both the formation and the removal of autophagosomes will ameliorate cardiac hypertrophy and HF [29].

Finally, increasing evidence consistently demonstrates that the calpain/calpastatin system, which includes a family of calcium-dependent, non-lysosomal cysteine proteases and their endogenous inhibitor calpastatin, is involved in the development of maladaptive hypertrophy triggered by numerous pathologic stimuli [38,39,40,41]. Chronic cardiac stress induces calpain overexpression and overactivation, resulting in the proteolysis of a broad spectrum of substrates, some of them with important functions as regulators of intracellular pathways classically associated with the development of hypertrophy. However, despite the solid preclinical evidence demonstrating that pharmacological calpain inhibition is feasible and may be an effective therapeutic intervention for treating hypertrophy [38,40,41], translation of these studies into the clinic is still at an early development stage. This review aims to summarize the most recent evidence supporting the contribution of calpains to cardiac hypertrophy and to overview the advances and limitations in the design of calpain inhibitors for their use in patients.

## 2. The Conventional Calpain/Calpastatin System

Calpains are a cysteine protease family directly activated by Ca^2+^ and regulated by their endogenous specific inhibitor calpastatin [42]. Until now, 15 isoforms have been described in humans [43]. Among them, the most ubiquitous and well-known calpain isoforms are calpain-1 and calpain-2, which along with calpastatin, conform to the conventional calpain/calpastatin system. Unconventional calpains are generally expressed in a tissue-specific manner and/or present a non-classical protein structure [44]. In the current review, the term calpain refers to calpain-1 and calpain-2 isoforms. They are both constituted by a large 80-kDa catalytic subunit (CAPN1 and CAPN2) and a common 30-kDa regulatory subunit CAPN4 [45]. The catalytic subunit comprises four major domains. Domain I (N-terminal anchor helix domain) contains a site for autolysis in response to Ca^2+^. Domain II (cysteine protease domain) is divided into two sub-domains, IIa and IIb, each containing one conserved Ca^2+^ binding site. This domain harbors the catalytic triad residues formed by Cys, His, and Asn. These two domains conform the most conserved region of the protein and define the calpain family [44]. Domain III (β-sandwich domain) is suggested to be the primary phospholipid binding site of calpain [46]. Domain IV (C-terminal domain) includes five penta-EF-hands (PEF) Ca^2+^-binding sequences [47]. The regulatory subunit is composed of two domains and also contains five PEF binding sequences [47]. Non-classical or tissue-specific members of the calpain family are often monomeric and lack both the N-terminus and the PEF domain.

Calpain-1 and calpain-2, also known as μ and m-calpain, are activated by in vitro micromolar and millimolar Ca^2+^ concentrations, respectively [42], which are above the physiological levels of Ca^2+^ found in live cells. However, different mechanisms, including their translocation to the cell membrane, recruitment in Ca^2+^ hotspots, and autoproteolysis, may contribute to reducing the minimal concentration of Ca^2+^ required for their activation in vivo [43].

Calpains have been commonly involved in all those physiological processes that are regulated by Ca^2+^, including embryonic development, cytoskeletal remodeling, cell cycle progression [48], cell spreading and migration [49], membrane repair [50], and platelet function [51]. In contrast to other major intracellular proteolytic components, such as proteasome and lysosomal proteases, calpains do not induce protein digestion but regulate protein functions through the limited proteolysis of their substrates [52]. The genetic disruption of calpain-1 or calpain-2 genes has been reported to drive different phenotypes, and while mice with genetic depletion of calpain-1 appear normal and are fertile [53], calpain-2 knockout mice die before the blastocyst stage [54]. These studies suggest that calpains 1 and 2 differ in their physiological functions and/or expression levels, at least during developmental stages.

It is well established that under pathological conditions resulting in the loss of Ca^2+^ homeostasis, calpains are overactivated and deregulated [55,56,57]. Transgenic mouse models with an altered calpain/calpastatin system and the use of calpain inhibitors consistently show that calpains play a key or contributory role in the pathology of a variety of cardiac disorders, including platelet aggregation, myocardial ischemia, and HF [58].

## 3. Common Murine Models of Hypertrophy

Many different preclinical models of HF have been adopted to study the development and progression of pathological cardiac hypertrophy, and a detailed description of the translational relevance and limitations of each one can be found in recent reviews [10,59,60,61]. The use of mice for this purpose has evident advantages, including their short generation times and the possibility of using genetically engineered models. Herein, we briefly describe those mouse models that have been used to examine the link between calpains and hypertrophy.

### 3.1. Transverse Aortic Constriction (TAC)

This model mimics the myocardial adaptations associated with hypertension and aortic stenosis, and it is achieved by the permanent constriction of the aortic arch [62,63]. The chronic left ventricular pressure overload generated triggers concentric cardiac hypertrophy, as well as diastolic dysfunction, and ultimately leads to HF with reduced ejection fraction (HFrEF). TAC is a well-established and reliable model to induce hypertrophy, and specific cardiac hypertrophic phenotypes can be generated by controlling the degree of constriction [64,65]. Recently, this model has been adapted to be less invasive [66] and to allow debanding, facilitating the study of reverse cardiac remodeling [67].

### 3.2. Pulmonary Artery Constriction (PAC)

Permanent constriction of the pulmonary artery generates a relevant model to study the right ventricular remodeling and dysfunction that occurs as a consequence of pulmonary artery hypertension [68]. The right ventricle (RV) initially adapts to pressure overloading via concentric hypertrophy. However, as pressure overload persists, it evolves into maladaptive failing RV with dilation, decreased EF, and impaired ventricular-arterial coupling.

### 3.3. Myocardial Infarction (MI)

Myocardial infarction can be induced by permanent left anterior descending coronary artery (LAD) ligation or transient LAD ligation (ischemia/reperfusion). Permanent LAD ligation resembles the clinical situation of a significant fraction of patients that suffer acute MI without timely revascularization [69]. Transient LAD ligation mimics MI followed by successful revascularization, which reflects the clinical course of the majority of ST-elevation myocardial infarction (STEMI) patients. Both, transient and permanent LAD ligation models reproduce features of human HFrEF. Although the severity of the remodeling process in the transient LAD ligation highly depends on the duration of ischemia and the extent of the area at risk, it is typically less severe than in the models of permanent LAD ligation [70]. Viable cardiomyocytes induce cardiac pressure and volume overload, increasing neurohumoral activation and ventricular wall stretch that induces cardiac hypertrophy in the remote area of the myocardium [71].

### 3.4. Angiotensin II Administration

This model mimics hypertension and the neurohumoral activation observed in patients with HF, which includes the elevation of angiotensin II levels due to the activation of the renin–angiotensin–aldosterone-system (RAAS) [72]. Mice chronically treated with angiotensin II show concentric hypertrophy accompanied by diastolic dysfunction, and thus, they reproduce features of human HF with preserved ejection fraction (HFpEF). Moreover, depending on the dose and the duration of the angiotensin II treatment, they can exhibit LV dilatation accompanied by reduced EF and simulate human HFrEF [73,74].

### 3.5. Isoproterenol Administration

Activation of the sympathetic nervous system is associated with increased levels of catecholamines, cardiac hypertrophy, and HF in humans [75,76]. Chronic administration of the synthetic β-adrenergic agonist isoproterenol represents the most widely used model to mimic sustained adrenergic stimulation, and the activation of multiple downstream signaling pathways results in cardiomyocyte concentric hypertrophy and fibrosis without hypertension [38,77]. As occurs with angiotensin II, isoproterenol effects vary depending on the mode of administration, the administered dose, and the duration of treatment.

### 3.6. Streptozotocin-Induced Diabetic Cardiomyopathy

Diabetic cardiomyopathy develops in diabetic patients, and it is defined by the existence of abnormal cardiac performance preceded by hypertrophy in the absence of other cardiac risk factors, such as coronary artery disease or hypertension [78]. The most common and best characterized preclinical model of diabetic cardiomyopathy is based on the administration of streptozotocin, an antibiotic that induces insulin-dependent diabetes mellitus by causing the destruction of pancreatic islet β-cells [79]. Streptozotocin-induced diabetic mice show many of the features found in human diabetic cardiomyopathy, including increased levels of natriuretic peptides [80], inflammatory markers, and eccentric cardiac hypertrophy [80].

## 4. Calpain Activation during Cardiac Hypertrophy

Calpain activity is tightly controlled by their specific endogenous calpain inhibitor calpastatin and by variations in the intracellular Ca^2+^ homeostasis [42]. The control of intracellular Ca^2+^ concentration is ultimately maintained through the activity of Na^+^/K^+^-ATPase in a highly energy-consuming process [81]. In those situations associated with reduced mitochondrial energetic production, as occurs during myocardial ischemia, loss of Ca^2+^ control induces an excessive and dysregulated activation of calpains. Exaggerated calpain activation in the ischemic myocardium results in altered contractility and cell death through the cleavage of numerous protein substrates involved in sarcolemmal structure, cellular contractility, mitochondrial function, and cellular signaling [82]. Although less characterized than in the context of ischemia, abnormal Ca^2+^ handling has also been described in hypertrophic cardiomyocytes isolated from models of pressure overload [83]. There is evidence of the impairment of SERCA and cytosolic Ca^2+^ overload in cardiomyocytes during the transition from adaptive cardiac hypertrophy to pathological cardiac hypertrophy [84,85]. These changes have been associated with an early reduction in the activity of Na^+^/K^+^-ATPase and an increase in late Na^+^ current [86], conditions that reduce the transarcolemmal Na^+^ gradient and favor Ca^2+^ entry through the reverse mode of the Na^+^/Ca^2+^ exchanger [60]. More recently, it has been described that the mechanosensitive ion channels Piezo 1 are upregulated in the hypertrophic myocardium and promote Ca^2+^ entry [87,88]. Conditional and cardiospecific deletion of Piezo1 reduced Ca^2+^ overload and calpain activity and blunted myocardial hypertrophy induced by isoproterenol and aortic constriction in mice [87].

In addition to Ca^2+^ dysregulation, the overexpression of calpain-1 and calpain-2 is a common feature of preclinical models of cardiac remodeling and HF, including MI with transient and permanent LAD ligation, TAC, and chronic administration of isoproterenol and angiotensin II [14,38,41,89,90,91], and it has also been confirmed in myocardial samples from patients with HF [41,57,90,92,93]. Although the mechanism underlying calpain overexpression during the development of hypertrophy remains to be established, it has been described that the conditional overexpression of calpain-1 in cardiomyocytes, or restricted to mitochondria, results in increased global calpain activity and heart remodeling even in the absence of significant variations in intracellular Ca^2+^ [94].

Finally, it has been suggested that the activation of the stress-activated serine/threonine kinase p38γ MAPK in conditions of pressure overload can phosphorylate calpastatin and reduce its inhibitory efficiency, resulting in increased calpain activity [95].

Globally, these studies demonstrate that calpain overactivation is a general trait of myocardial hypertrophy, regardless of the triggering stimulus, and it is a consequence of calpain overexpression, altered cellular Ca^2+^ dynamics, and/or reduced calpastatin inhibitory capacity.

## 5. Evidence Supporting the Contribution of Calpains to Cardiac Hypertrophy

Mounting evidence obtained from studies using genetic models with altered calpain systems or calpain inhibitors demonstrates that calpains are involved in cardiac hypertrophy triggered by a variety of chronic pathologic stimuli (see Table 1 for details). In mice, the restriction of calpain-1 and calpain-2 activities by using a cardiomyocyte-specific deletion of the common subunit Capn4 or by overexpressing calpastatin reduced adverse-post-infarction remodeling and mortality [89,96,97]. Conversely, the genetic deletion of calpastatin increased calpain activity and had the opposite effect in the same experimental model [98]. In addition to post-infarction remodeling, the suppression of Capn4 or the constitutive overexpression of calpastatin attenuated cardiomyocyte hypertrophy and cardiac dysfunction induced by chronic angiotensin II treatment [39,99]. Providing further evidence of calpain contribution to myocardial remodeling, the inhibition of calpain by deletion of Capn4 prevented hypertrophy in a model of pulmonary hypertension [100], and calpastatin overexpression reduced myocardial hypertrophy and fibrosis in a mouse model of type 1 diabetes [101]. Besides the use of transgenic models, some studies have explored whether the pharmacological inhibition of calpains prevents cardiac hypertrophy. Among them, our group has shown that the sustained oral administration of the calpain inhibitor SNJ1945 in a model of MI with transient LAD ligation attenuates adverse post-infarction remodeling independently of its cardioprotective effects during the acute phase of reperfusion. These effects were associated with reduced hypertrophic, fibrotic, and inflammatory responses in the non-infarcted myocardium [40]. More recently, the same compound was also proven to be effective in preventing cardiac hypertrophy induced by chronic administration of isoproterenol to rats and mice [38]. Furthermore, Wang et al., showed that inhibition of calpain activity by daily intraperitoneal administration of MDL-28170 protected against pathological hypertrophy and cardiac dysfunction in multiple rodent models of HF, including MI, TAC, and angiotensin II treatment [41]. It is important to note that no signals of toxicity due to the long-term inhibition of calpains were reported in these studies.

## 6. Pro-Hypertrophic Pathways Modulated by Calpains

Development of pathologic cardiac hypertrophy involves a vast network of receptors, signaling pathways, and effector proteins that result in the activation of transcription factors, which, in turn, activate pro-hypertrophic gene expression programs. Excellent reviews covering the general topic have been written [112,113]. Here, instead, we focus on those signaling pathways that have been proposed to be regulated by calpains (Figure 1).

### 6.1. Calmodulin/NFAT Pathway

In addition to calpains, intracellular Ca^2+^ dysregulation is secondary to chronic stress, which also results in the activation of other Ca^2+^-dependent enzymes, such as calcineurin. Calcineurin is a Ca^2+^ and calmodulin sensitive phosphatase which, among other functions, plays a master regulatory role in the hypertrophic response of cardiomyocytes to a chronic stimulus by activating the transcription factor NFAT [114]. Further, activation of the calcineurin/NFAT pathway has been proposed to participate in pathological but not physiological forms of cardiac hypertrophy [115]. Interestingly, different studies suggest that calpain activation can modulate this signaling cascade by acting at different levels. Angiotensin II stimulation of cardiomyocytes has been shown to produce calpain-dependent proteolysis of the autoinhibitory domain of calcineurin resulting in a constitutive nuclear form, which remains active even after removal of the hypertrophic stimulus [116]. In addition, in vitro studies propose that calpain-1 may also activate calcineurin by cleaving the calcineurin-binding domain of the endogenous calcineurin inhibitor cain/cabin [117]. Consistent with this scenario, the attenuation of calpain activity by constitutive calpastatin overexpression or Capn4 genetic deletion reduced the activation of the NFAT pathway and attenuated myocardial hypertrophy in a mouse model of type 1 diabetes [101]. By contrast, inhibition of calpain by using the same calpastatin overexpressing mouse strain prevented cardiac hypertrophy induced by chronic infusion of angiotensin II through a mechanism independent of NFAT activation but dependent on the translocation to the nucleus of NF-κB [39].

### 6.2. NF-κB Activation

A solid body of evidence supports a critical role of the nuclear factor NF-κB in cardiac hypertrophy induced by a wide variety of chronic pathologic stimuli. Targeted disruption of the p50 NF-κB subunit reduced cardiomyocyte hypertrophy and improved cardiac function after MI [118], while deletion of its c-Rel subunit ameliorated cardiac hypertrophy in response to chronic infusion of angiotensin II [118,119]. In line with these studies, the administration of the NF-κB inhibitor, PDTC, reduced cardiac hypertrophy resulting from angiotensin II [120] or isoproterenol treatments [121].

Accumulating evidence obtained by different groups convincingly demonstrates that calpain overactivation enhances NF-κB activity. Calpain inhibition by genetic deletion of Capn4 or calpastatin overexpression prevents the nuclear translocation of the p65 NF-κB subunit and attenuates hypertrophy induced by myocardial ischemia and angiotensin II [39,89]. Similar results have been obtained by using calpain inhibitors in models of MI with transient LAD ligation or isoproterenol administration [38,40].

NF-κB is sequestered in the cytoplasm by the interaction with its inhibitory protein IκBα. The expression of a mutant IκBα that acts as a super-repressor of NF-κB in transgenic mice attenuates hypertrophy induced by isoproterenol or angiotensin II infusion [122], pointing to IκBα degradation as a necessary step for the nuclear translocation of the p65 NF-κB subunit and the progress of pathological hypertrophy. Although canonical IκBα proteolysis involves the ubiquitin-proteasome pathway, IκBα is a well-known calpain substrate [123] and its calpain-dependent cleavage promotes hypertrophy in response to MI [40], angiotensin II [122], adrenergic stimulation [38], and in streptozotocin-induced diabetic rats [124]. Altogether, these studies demonstrate that calpains contribute to cardiomyocyte hypertrophy at least in part by activating NF-κB through the direct proteolysis of IκBα.

### 6.3. GRK2 Upregulation

Angiotensin II, endothelin-1, phenylephrine, and isoproterenol are well documented, showing cardiac hypertrophy by activating pathways linked to G protein-coupled receptors (GPCRs). G protein-coupled receptor kinases (GRKs) are key modulators of GPCRs in physiological and pathological conditions and have attracted a lot of attention primarily for their role in regulating β-adrenergic receptors in the context of cardiac contraction [125,126]. In addition, different groups provide evidence of non-canonical roles of the GRK2 isoform, including its ability to modulate cardiac hypertrophy [127]. GRK2 is upregulated in patients with HF and preclinical models of chronic stress of either hypertensive or ischemic origin [38,128]. In H9c2 cells, a rat myoblast cell line, the overexpression of GRK2 is enough to elicit a hypertrophic response [129], while its genetic deletion attenuates hypertrophy induced by TAC or isoproterenol administration [38,128]. A recent study from our group demonstrates that calpain activation in response to chronic isoproterenol administration promotes the overexpression of GRK2 by mechanisms affecting both its stability and transcription [38]. Isoproterenol treatment induces a calpain-dependent decrease of cardiac MDM2 levels, the major E3 ligase implicated in the ubiquitination and degradation of GRK2 [130], thus enhancing GRK2 stability. Moreover, the GRK2 promoter sequence has several canonical binding sites for NF-κB. Isoproterenol-induced calpain activation cleavages IκBα, leading to NF-κB translocation, which enhances the transcriptional activity of the GRK2 promoter. Further, it has been proposed that GRK2 can also phosphorylate IκBα, favoring its proteasomal degradation and the subsequent activation of NF-κB [129], suggesting that myocardial GRK2 and NF-κB co-regulate each other to trigger hypertrophic gene transcriptional activation. Remarkably, chronic administration of an oral calpain inhibitor prevented isoproterenol-dependent GRK2 upregulation, while hemizygous GRK2 mice showed attenuated myocardial hypertrophy. Overall, these studies strongly suggest that the calpain-dependent modulation of the MDM2/GRK2 axis is a relevant event in cardiac hypertrophy downstream calpain overactivation.

### 6.4. Junctophilin-2 Cleavage

Recently, the calpain-dependent proteolysis of junctophilin-2 (JPH2) has emerged as a novel mechanism involved in the regulation of cardiomyocyte growth. Junctophilin-2, a structural protein connecting T-tubules and the sarcoplasmic reticulum, is essential for maintaining normal T-tubule organization and an efficient excitation–contraction coupling in adult cardiomyocytes [131]. Different studies demonstrate that calpain activation induced by several models of cardiac stress, including MI with permanent LAD occlusion, TAC, and isoproterenol infusion, results in the cleavage of JPH2 and the disruption of the contractile machinery, driving HF progression [41,132,133]. The correlation between the reduction of JPH2 levels and increased calpain activity has also been confirmed in failing human hearts [41]. In addition to its effects on contractility, it has been recently suggested that calpain-1 cleaves JPH2 at a conserved R565/T566 site and the resulting JPH2 N-terminal fragment (JPH2NT) translocates to the nucleus, where it acts as a stress-adaptive transcription regulator through Mef2 gene repression [134]. Supporting this role of JPH2NT, the transgenic overexpression of JPH2NT attenuated pathological remodeling in response to TAC, while genetic mice with a loss of function of JP2NT exacerbated hypertrophy and cardiac dysfunction. More recently, the nuclear localization of a novel C-terminal fragment (JPH2CT) generated by the calpain-2-dependent cleavage of JPH2 in preclinical models of pressure overload and adrenergic stimulation has been described and also observed in ventricular samples from HF patients [90]. Most interesting, however, is that contrary to the effects of JPH2NT, the blockade of nuclear localization of JPH2CT protected cardiomyocytes from isoproterenol-induced hypertrophy. Considering that these two fragments modulate cardiomyocyte growth in apparently opposite directions and that each one is generated by a specific calpain isoform, it can be speculated that the intracellular Ca^2+^ concentration will determine the predominant JPH2 fragment. According to this hypothesis, JPH2CT, resulting from calpain-2, may have a preferential contribution in pathological conditions, while JPH2NT, resulting from the calpain-1 activity, in physiological or compensatory conditions. More recently, it has been suggested that calpain-2 can also cleave JPH2 at the same site as calpain-1, although with less efficacy [135].

## 7. Calpain Contribution to the Progression of Pathologic Hypertrophy

In addition to their direct effect on the genesis of hypertrophy, calpains have also been proposed to participate in the progression of hypertrophy to HF.

### 7.1. Proteolysis of Myosin Light Chain Kinase

Evidence from studies conducted in a knockout myosin light chain kinase (MLCK) mouse model and a cardiac-specific overexpressing MLCK transgenic mouse model suggests an important role for cardiac myosin light chain (MLC) phosphorylation in the evolution of cardiac hypertrophy to HF [136,137]. Importantly, it has been described that β-adrenergic stimulation of neonatal cardiomyocytes and pressure overload produced by TAC induce a calpain-dependent proteolysis of MLCK [138].

### 7.2. Mitochondrial Damage

Although traditionally considered cytoplasmic proteases, different studies have reported that calpains 1, 2, 4, and 10 are also found in mitochondria [139,140,141]. Accumulating evidence, recently reviewed by Zhang et al. [142], suggests that both cytosolic and mitochondrial calpain dysregulation may induce mitochondrial damage. Cardiac energy deprivation and increased ROS production resulting from mitochondrial dysfunction due to calpain overactivation during hypertrophy may promote the transition to decompensated hypertrophy and HF [143].

Calpain overactivation was initially linked to the induction of the mitochondrial-dependent apoptotic program by activating the pro-apoptotic factors Bid [144,145] and AIF [140]. However, the relevance of apoptotic cardiomyocyte death in the context of acute reperfusion injury and cardiac remodeling has been questioned due to the repression of the canonical caspase pathway in post-mitotic cardiomyocytes [146,147].

Instead, more recent studies propose that mitochondrial calpains contribute to the direct damage of the electron transporter chain (ETC) by targeting the NDUFS7 [148] and ND6 [141] subunits of complex I. Mitochondrial calpains have also been involved in the disruption of the mitochondrial FoF1 ATP synthase through the proteolysis of its ATP5A1 subunit [149]. More recently, by using transgenic mice with calpain-1 upregulation restricted to cardiomyocyte mitochondria, the same group has demonstrated a causal association between calpain-mediated cleavage of ATP5A1 and ROS generation, mPTP opening, and cell death [150].

Finally, mitochondrial calpains have been suggested to alter mitochondrial dynamics [151,152]. Mitofusin 2, which plays a central role in mitochondrial fusion, has been identified as a direct substrate of calpains [153]. More recently, cardiac-specific downregulation of OPA1, a dynamin-related GTPase protein involved in mitochondrial fusion [154] and in maintaining mitochondrial cristae structure [155], has been associated with mitophagy inhibition and enhanced cardiomyocyte death in the setting of myocardial infarction [156]. In a recent study, calpastatin overexpression in mice subjected to myocardial infarction prevented OPA1 degradation and improved mitochondrial fusion and mitophagy [157]. However, whether calpain directly targets OPA1 or modulates OPA1 expression through an indirect mechanism remains to be elucidated. Furthermore, calpain inhibition has been shown to prevent beclin-1 cleavage, a key component of the autophagy pathway required to form autophagosomes, and improved mitophagy in isolated hearts subjected to transient ischemia [148]. Altogether, these studies suggest that calpain overactivation negatively modulates mitophagy by acting at multiple levels.

## 8. Pharmacological Inhibition of Calpains

Despite the accumulating experimental evidence supporting the contribution of calpains to the development of hypertrophy, myocardial remodeling, and its progression to HF, no clinical trials have explored the pharmacological inhibition of calpains as a therapeutic strategy yet. The main reason for this is related to the limitations of most of the available calpain inhibitors, which involve low selectivity, limited membrane permeability, and reduced water solubility and metabolic stability [52]. However, the research in the development of novel calpain inhibitors has been greatly benefited from the increasing evidence demonstrating the contribution of calpains to pathologic processes involved in adverse myocardial remodeling other than hypertrophy, including fibrosis and inflammation [158], and non-cardiac pathologies, such as neurodegenerative disorders [146], ophthalmic diseases [159], myopathies [160], and cancer [161] (extensively discussed in previous reviews [162]).

E-64 was the first calpain inhibitor used in rats to suggest the involvement of calpains in cardiac hypertrophy [104]. E-64 and leupeptin constitute the first generation of calpain inhibitors, and their structure contains a peptidyl backbone and an electrophilic warhead that covalently interacts with the active site cysteine of calpain. However, although extensively used, these molecules show limited specificity for calpains and low membrane permeability [163]. The pharmacological properties of leupeptin were improved by substituting its amino-terminal for a hydrophobic cap group [163]. The inhibition of calpain activity using one of these synthetic leupeptin derivatives, MDL-28170 (calpain inhibitor III), has demonstrated efficacy against pathological hypertrophy and cardiac dysfunction in multiple rodent models of HF, including MI, TAC, and chronic isoproterenol infusion [41]. However, these compounds show poor drug-like properties due to insufficient bioavailability and unfavorable pharmacokinetics, and, therefore, their progression into the clinic has been excluded. Advances in the design of new peptidomimetic calpain inhibitors have provided new molecules with improved water solubility and metabolic stability over previous inhibitors. Among them, and deriving from the benzoylalanine-derived ketoamide calpain inhibitor A-705253, A-953227 showed potent calpain inhibitory properties combined with high selectivity versus related cysteine protease cathepsins, other proteases, and receptors and was effective in reducing infarct size in an in vivo pig model of IR [164]. However, the short effective half-life and low bioavailability caused by the instability against carbonyl reductases led to the development of a more metabolic stable derivative Alicapistat (ABT-957) [165]. Alicapistat reached a phase I clinical study that analyzes its safety and pharmacological properties for the treatment of Alzheimer’s disease [166]. However, although in preclinical models, alicapistat demonstrated efficacy with respect to the prevention of NMDA-induced neurodegeneration, it failed to induce any measurable hemodynamic effect in humans. This negative result was attributable to the use of an inadequate concentration and suggested a moderate inhibitory potency. The ketoamine derivative SNJ-1945, produced by Senju Pharmaceutical [167,168], shows an appropriate pharmacological profile, and its chronic oral administration was effective in preventing calpain activation and attenuating cardiomyocyte hypertrophy and cardiac dysfunction in a mouse model of transient LAD occlusion [40]. More recently, these favorable effects of SNJ-1945 have been confirmed in a model of hypertrophy induced by chronic isoproterenol administration [38]. Currently, a phase IIa clinical trial designed to test the efficacy and safety of the oral administration of SNJ-1945 in patients with non-arteritic retinal artery occlusion is in progress (jRTC2021190013).

The major limitation of calpain inhibitors is still their limited specificity for calpains over other cysteine proteases, mainly caused by the highly conserved active site among this type of proteases. A different approach aimed at increasing the specificity for calpains in the design of molecules that induce the allosteric inhibition of the enzyme by binding to other positions than the catalytic site. One of these allosteric inhibitors, PD150606, which is supposed to bind to the Ca^2+^ binding site of calpain, was effective in attenuating the development of hypertrophy in isolated cardiomyocytes treated with isoproterenol [169] or angiotensin II [170]. It is important to mention that a PD150606 derivative, PD151746, is proposed to be more effective in inhibiting calpain-1 than calpain-2 [171]. Considering that these two main calpain isoforms may display some differences in their substrate preference [172] and biological function [173,174], this type of compound opens the door to the development of new isoform-selective inhibitors.

Calpains also seem to show differences in their regulation depending on their intracellular localization. It has been proposed that the activity of mitochondrial but not cytosolic calpains is regulated by its binding to chaperons (ERp57 for calpain-1 and Grp75 for calpain-2) [175], and the use of peptides that blocks their interaction inhibits the mitochondrial activity of calpain-1 in a specific manner [176]. Considering that mitochondria play a critical role in the development of hypertrophy [177,178] and that mitochondrial calpain-1 and/or calpain-2 increase in response to pathological stress associated with the development of HF [150,179], the potential therapeutic benefits of the compounds selectively addressed to mitochondrial calpains deserve further investigation.

## 9. Conclusions

Chronic myocardial stress invariably results in the overexpression and overactivation of calpains. Several preclinical studies using transgenic models with an altered calpain/calpastatin system and pharmacological inhibitors support the contribution of calpains to the development of cardiac hypertrophy and its progression to adverse remodeling and cardiac dysfunction. However, although the evidence reviewed herein convincingly suggests that calpain inhibition is an attractive novel therapeutic strategy, the clinical use of currently available calpain inhibitors has been hampered by their low selectivity [163] and inappropriate pharmacologic profile. Therefore, most of the calpain inhibitors tested in preclinical studies did not meet the requirements to be candidates for use in patients. Fortunately, new advances can overcome these limitations. On the one hand, there is evidence for the involvement of both cathepsins and calpains in different pathophysiological conditions, including hypertrophy [180,181,182]. In fact, the term “calpain−cathepsin” hypothesis has been used to describe the coordinated and dysregulated proteolytic actions of calpain-1 and cathepsin-B, causing neurodegeneration in multiple disorders [183]. Although this hypothesis needs to be confirmed in the context of cardiac hypertrophy, it is reasonable to question whether the design of pure specific calpain inhibitors is the best strategy for the achievement of optimal clinical results [184]. On the other hand, the increasing number of studies supporting the involvement of calpains in other non-cardiac pathologies has served to promote the development of novel calpain inhibitors with more favorable pharmacologic profiles, and new molecules are constantly being described. Meanwhile, the safety and pharmacokinetics of SNJ-1945 are currently being tested in a phase II clinical trial (jRTC2021190013). More recently, a novel calpain inhibitor, BLD-2660 (BLADE Therapeutics), has been approved for a phase II clinical trial that evaluates its safety and antiviral activity in hospitalized subjects with COVID-19 (NCT04334460).

## Figures and Tables

**Figure 1 ijms-23-04103-f001:**
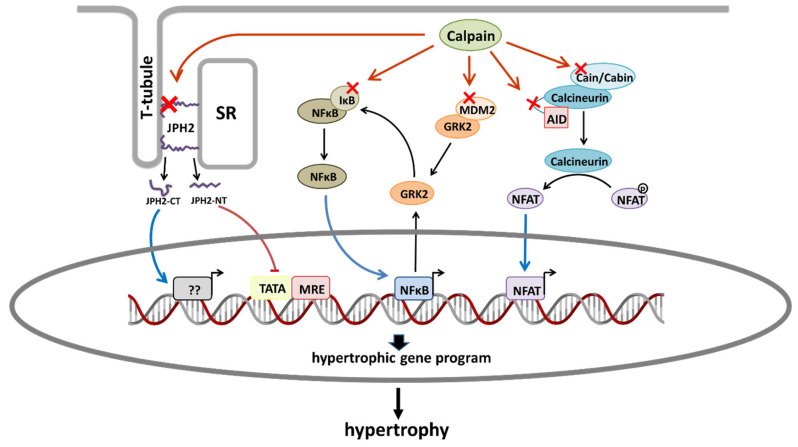
Schematic diagram showing the main proposed mechanisms by which calpains promote cardiac hypertrophy. Red crosses indicate calpain substrates that are involved in hypertrophic signaling pathways. From left to right: Calpain-2-dependent proteolysis of JPH2 generates a JPH2-CT fragment that translocates to the nucleus and favors hypertrophy. Calpain-1-dependent proteolysis of JPH2 produces a JPH2-NT fragment that acts as a stress-adaptive transcription regulator preventing hypertrophy. Calpain-dependent degradation of IkBα activates NFκB. Calpain activity promotes the upregulation of GRK2 by mechanisms affecting both its stability (degradation of MDM2) and transcription (activation of NFκB). GRK2 overexpression phosphorylates IκBα promoting its proteosomal degradation and the subsequent activation of NFκB. Proteolysis of cain/cabin or calcineurin AID induces the activation of NFAT. AID, autoinhibitory domain; JPH2, junctophilin 2; JPH2-CT, junctophilin 2 C-terminal fragment; JPH2-NT; SR, sarcoplasmic reticulum.

**Table 1 ijms-23-04103-t001:** Selected studies suggesting the calpain contribution to cardiac hypertrophy in different preclinical models of chronic stress and in patients with heart failure.

Species	Model	Calpain Activity/ Expression	Inhibitor/Transgenic	Hallmarks	Reference
Mouse	Ischemia TAC Isoproterenol	↑Calpain activity ↑CAPN1	MDL-28170	MDL: ↓Hypertrophy, ↑LV contractile function, ↓Fibrosis	[41]
Mouse			Cardiomyocyte-conditional CAPN1 overexpression	↑Hypertrophy ↓LV contractile function	[41]
Mouse	TAC	↑Calpain activity ↑CAPN2			[90]
Mouse	Diabetic cardiomyopathy	↑Calpain activity	MDL-28170 Cardiomyocyte-specific CAPN4 KO CAST overexpression	↓Hypertrophy ↓Fibrosis	[101]
Mouse	Angiotensin II	↑Calpain activity	CAST overexpression	↓Hypertrophy ↓Perivascular inflammation, fibrosis and recruitment of mononuclear cells	[39]
Mouse	Ischemia	↑Calpain activity ↑CAPN1 and CAPN2	CAST overexpression	↓Hypertrophy ↓Fibrosis	[89]
Mouse	Diabetic cardiomyopathy	↑Calpain activity	Cardiomyocyte-specific CAPN1 KO	↓Hypertrophy ↓Fibrosis	[102]
Mouse	TAC	≈CAPN1 and CAPN2	Calpeptin	↓Programmed cell death	[103]
Rat	Isoproterenol	↑Calpain activity ↑CAPN1 and CAPN2	SNJ-1945	↓Hypertrophy	[38]
Rat	Isoproterenol	↑Calpain activity	E64c	↓Hypertrophy	[104]
Rat	Ischemia	↑Calpain activity	Calpain inhibitor XII	≈Hypertrophy	[105]
Rat	Ischemia	↑Calpain activity ↑CAPN1		↑Hypertrophy	[106]
Rat	Ischemia	↑Calpain activity	CAL 9961	↓Hypertrophy	[107]
Rat	Ischemia	↑Calpain activity ↑CAPN1 and CAPN2		↑Hypertrophy	[108]
Rat	Ischemia	↑Calpain activity ↑CAPN1 and CAPN2		↑Hypertrophy ↑Fibrosis	[109]
Rat	Ischemia/ Reperfusion	↑Calpain activity ↑CAPN1 and CAPN2	SNJ-1945	↓Hypertrophy ↓Fibrosis ↓Inflammation	[40]
Rat	TAC	↑Calpain activity ↑CAPN1		↑Hypertrophy ↑Fibrosis	[14]
Pig	Ischemia	↑Calpain activity	MDL28170	↓Fibrosis	[110]
Rat	DOCA-salt	↑Calpain activity		↑Hypertrophy	[111]
Rat	Ischemia Angiotensin II	↑CAPN1 and CAPN2			[91]
Human	Valvular heart disease	↑Calpain activity ↑CAPN1			[92]
Human	Ischemic or dilated cardiomyopathy	↑Calpain activity ↑CAPN1			[41]
Human	End-stage heart failure	↑Calpain activity ↑CAPN2			[90]

CAPN1: calpain 1; CAPN2: calpain 2; CAST: calpastatin; TAC: transverse aortic constriction.

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
