# Peer review of "Calpains as Potential Therapeutic Targets for Myocardial Hypertrophy"

_ijms, 2022, doi:10.3390/ijms23084103_

Round 1

Reviewer 1 Report

This manuscript summarized and described a role of calpain as potential therapeutic target for myocardial hypertrophy. For better reading and benefits to the readers, following concerns were highlighted.

  1. Current form of manuscript is of difficulty in reading and understanding. Please re-organize the whole structures of manuscript in a well-categorized form, including head and section numbers.
  2. Clinical situations of myocardial hypertrophy and relevant causing diseases and complications should be provided for a general understanding of backgrounds.
  3. Action targets and subcellular distribution are key components of calpain biological impacts, such as membrane translocation, mitochondrial targeting, autophagy molecules, and apoptosis molecules. In additional to so called signaling concerns, the relevant information is crucial for the whole picture of calpain implications.
  4. Figure 1 highlights a common signaling to the nuclear transcription factors. The effectors crucial to the development of myocardial hypertrophy under the control of those proposed transcriptional programs are of interests. Please add the candidates to the figure.

Reviewer 2 Report

The authors have described about the Calpains and its role in cardiac hypertrophy in this review. Moreover, they have discussed about role of the calpains inhibitors. The review is well written and the references are up to date. My comments are provided below

  1. In the introduction section, the authors should provide a summary about different members of Calpain family. In this respect, they should describe the domain organization of the Calpain1 and 2.
  2. The authors should describe about the different mutations in Calpain 1 and 2 responsible for cardiac hypertrophy.
  3. The authors did not describe about the Calpain1 and 2 knock out mice phenotype.
  4. It would be better if the authors could provide a table summarizing the role and clinical trial stages of Calpain inhibitors.

Round 2

Reviewer 1 Report

There is no additional comment.

Reviewer 2 Report

The authors have addressed all my concerns in the revised text. I support the publication of the manuscript.